# Synergistic Effects of Silicate-Platelet Supporting Ag and ZnO, Offering High Antibacterial Activity and Low Cytotoxicity

**DOI:** 10.3390/ijms24087024

**Published:** 2023-04-10

**Authors:** Chih-Hao Chang, Li-Hui Tsai, Yi-Chen Lee, Wei-Cheng Yao, Jiang-Jen Lin

**Affiliations:** 1Department of Orthopedics, National Taiwan University Hospital Jin-Shan Branch, New Taipei City 20844, Taiwan; 2Department of Orthopedics, National Taiwan University Hospital and National Taiwan University College of Medicine, Taipei 100, Taiwan; 3Department of Biomedical Engineering, National Taiwan University, Taipei 100, Taiwan; d07528009@ntu.edu.tw; 4Institute of Polymer Science and Engineering, National Taiwan University, Taipei 10617, Taiwan; 5Department of Anesthesiology and Pain Medicine, Min-Sheng General Hospital, Taoyuan 330, Taiwan

**Keywords:** nano-silicate platelet, silver nanoparticles, zinc oxide nanoparticle, antibacterial, cytotoxicity

## Abstract

Silver nanoparticles (AgNPs) are remarkably able to eliminate microorganisms, but induce cytotoxicity in mammalian cells, and zinc oxide nanoparticles (ZnONPs) are considered to have a wide bactericidal effect with weak cytotoxicity. In this study, both zinc oxide nanoparticles and silver nanoparticles were co-synthesized on a nano-silicate platelet (NSP) to prepare a hybrid of AgNP/ZnONP/NSP. Ultraviolet–visible spectroscopy (UV-Vis), X-ray diffraction (XRD), and transmission electron microscopy (TEM) were used to characterize the formation of nanoparticles on the NSP. Synthesized ZnONP/NSP (ZnONP on NSP) was confirmed by the absorption peaks on UV-Vis and XRD. AgNP synthesized on ZnONP/NSP was also characterized by UV-Vis, and ZnONP/NSP showed no interference with synthesis. The images of TEM demonstrated that NSP provides physical support for the growth of nanoparticles and could prevent the inherent aggregation of ZnONP. In antibacterial tests, AgNP/ZnONP/NSP exhibited more efficacy against Staphylococcus aureus (*S. aureus*) than ZnONP/NSP (ZnONP was synthesized on NSP) and AgNP/NSP (AgNP was synthesized on NSP). In cell culture tests, 1/10/99 (weight ratio) of AgNP/ZnONP/NSP exhibited low cytotoxicity for mammalian cells (>100 ppm). Therefore, AgNP/ZnONP/NSP, containing both AgNP and ZnONP, with both strong antibacterial qualities and low cytotoxicity, showed potentially advantageous medical utilizations due to its antibacterial properties.

## 1. Introduction

Nanomaterials with a geometric size at the nanometer scale have been developed and are widely used in biotechnology [1,2], electronics [3], catalysis [4], and cosmetics [5]. Both the applications and practical functions of nanomaterials are generally related to their chemical compositions, dimensional sizes, geometric shapes, and surface/volume ratios [3]. Compared to their micro-size counterparts, metallic nanoparticles (NPs), such as zinc oxide NPs (ZnONP) and silver nanoparticles (AgNP), exhibit unique chemical properties, mechanisms, optics, electricity, and a higher surface-to-volume ratio. ZnONPs have been applied as bactericides, drug delivery vehicles, and bioimaging agents [6,7,8]; while AgNPs have also been used in biosensors and antibacterials [9,10,11].

Metal-containing NPs prepared by inorganic salt chemical reduction processes frequently require organic stabilizers, emulsifiers, and polymeric surfactants in an aqueous solution [12,13,14]. Water-soluble polymers, such as poly(vinyl alcohol) [15] and poly(vinylpyrrolidone) [16], were employed to stabilize the particles generated in an aqueous solution by preventing aggregation. In the preparation of ZnONPs, Tween 80 surfactants and hydrophilic poly(ethylene glycol) (PEG) are used as stabilizers to control particle aggregation [17,18,19,20]. However, the reactivity of these particles might be reduced, and side effects might be generated due to the wrapping of these organic stabilizers [21,22]. To eliminate the involvement of organic components, recent studies have utilized inorganic nanoscale materials to support metal-containing NPs [23,24,25].

The clay minerals are characterized as nontoxic and biocompatible silicates with a layered structure. The properties of clay minerals, such as high surface and water dispersibility, have been studied and applied to metal composites [26]. Among these silicate minerals, sodium montmorillonite (Na^+^-MMT) has a structure containing multilayered aluminosilicate sheets comprising exchangeable metal counter-ions. Na^+^-MMT, a bentonite clay, with 1.20 mequiv/g of cationic exchange capacity (CEC) or exchangeable Na^+^ capacity, is generally composed of 2/1 layered silicate/aluminum oxides, geometrically arranged in two tetrahedral sheets sandwiching an edge-shared octahedral sheet [27,28]. Previous studies have developed the method of layered exfoliation to create a nano-silicate platelet (NSP) with the geometric dimensions of approximately 80 × 80 × 1 nm^3^ [29,30]. The one-nanometer-thin dimension of NSP was characterized to have a high surface area at ca. 750 m^2^/g and 18,000 ions/platelet surface charges. NSPs have been revealed to restrain the proliferation and growth of bacteria in previous studies. The surface charge of NSPs, attributed to SiO-Na^+^, results in significant antimicrobial activities; platelets adhere to the surface of microbes due to their physical capturing mechanisms [31,32]. Moreover, surfactant-modified NSP has exhibited broad antiviral activities [33] and the potential to treat gray mold disease in strawberries [34]. Since NSP has demonstrated its safety with low cytotoxicity and genotoxicity [35], its use supporting nanoparticles, such as silver [36,37,38] and magnetic iron oxide [39], have been investigated for potential applications in the medical field.

The combination of different nanomaterials has shown greater potential due to the coupling of individual properties. The class of nanohybrids includes multi-metallic compositions, biomolecule–nanoparticle conjugate, polymer–nanoparticle, and other composites [40,41]. Hybrids can be further modified for diversified applications, including enhancing antibacterial activity and mitigating toxicity [42]. Silver has been applied in food protection and wound treatment, such as burning, for several years [43,44]. Numerous silver compounds, such as silver oxide, silver powder, silver chloride, and silver nitrate, are used as antimicrobial materials [45,46,47]. However, the issue of toxicity should be considered when using silver-based compounds as antimicrobial agents due to the cytotoxicity and genotoxicity in human cells shown in the previous study [48]. In our earlier works, NSP was hybridized with AgNP. The “naked” AgNPs were supported on the surface of NSP without organic stabilizers [49]. This AgNP/NSP (AgNP was synthesized on NSP) demonstrated antimicrobial activities against Gram-positive and Gram-negative bacteria. The particle size of AgNP and the antibacterial activity of AgNP/NSP were affected by the weight ratio of Ag and NSP [50]. In addition, AgNP/NSP demonstrated low cytotoxicity and further promoted wound healing [37]. Although the mitigation of Ag toxicity can be achieved by impeding AgNP’s penetration into cells through the tight tethering between AgNP and NSP [37,51] or by a polymer stabilizer reducing the exposure of cells to AgNPs [52], the uses of Ag-related products still elicited severe concerns regarding the toxicity toward mammalians and the environment [53]. Moreover, ZnONP has shown much lower cytotoxicity than AgNP toward mammalian cells [54,55] and higher antibacterial efficacy than CuO and Fe_2_O_3_ nanoparticles [56]. ZnONP, which is non-toxic in microorganisms and biocompatible in human cells, has been widely used as an antibacterial agent due to its antibacterial activities, for example, in food packaging, to inhibit the growth of foodborne pathogens. Four antibacterial mechanisms of ZnONP were proposed: contact with cell walls and ZnONP, destructing the integrity of cells, Zn^2+^ ion liberation, and reactive oxygen species (ROS) formation [57]. Hence, ZnONP was employed to hybridize with NSP (ZnONP/NSP) by the reduction and coprecipitation of aqueous Zn^2+^ salts on the surface of NSP. The bi-component AgNP/ZnONP/NSP (AgNP and ZnONP synthesized on NSP) was then prepared. The synthesis and characterization of AgNP/ZnONP/NSP were reported. The antimicrobial efficacies and safety issues were also revealed.

In the present study, AgNP/ZnONP/NSP was prepared using a two-step process, meaning that ZnONP/NSP was initially formed by the in situ dehydration of Zn(OH)_2_ on NSP, and AgNP was then deposited in the presence of the reducing agent (NaBH_4_) and AgNO_3_ solution. X-ray diffraction (XRD) was utilized to confirm the synthesis of ZnONP on NSP, and NSP seemed to prevent the self-aggregation of ZnONP. AgNP was characterized by ultraviolet–visible spectroscopy (UV-Vis) to ensure the formation of AgNP/ZnONP/NSP. A transmission electron microscope (TEM) was used to catch the morphology and determine the diameter of ZnONP, ZnONP/NSP, and AgNP/ZnONP/NSP. AgNP/ZnONP/NSP (weight ratio: 1/10/99) demonstrated low cytotoxicity when incubated with fibroblasts, based on ISO 10993-5. In addition, AgNP/ZnONP/NSP showed much stronger antibacterial activities against *S. aureus* than ZnONP/NSP and AgNP/NSP. The hybrid, AgNP/ZnONP/NSP, has shown high potential in clinical applications and the food industry.

## 2. Results and Discussion

### 2.1. Synthesis of ZnONP/NSP Hybrids

ZnONP/NSP hybrids were prepared by the ionic transfer of zinc ions (Zn^2+^) from a zinc acetate to a sodium ion (Na^+^) on the NSP. During the process, Zn(OH)_2_ was formed under alkaline conditions by the addition of NaOH. Afterward, ZnONP was in situ produced in the presence of NSP by the dehydration of Zn(OH)_2_ at the elevated temperature. The process of particles being synthesized on silicate platelets, which provided the support and the inorganic agent, is shown in Figure 1b. The synthesized ZnONP/NSP hybrids with the composition weight ratios of 1/99, 7/93, 15/85, and 30/70 were characterized by UV-Vis spectroscopy, as shown in Figure 1. The peak in UV-Vis absorption at 380 nm was undetectable until the ZnONP/NSP was shown at 7/93. Hybrids with different weight ratios of 7/93, 15/85, and 30/70 were further characterized by XRD analysis, as shown in Figure 2. By comparing the XRD patterns of the synthesized hybrids with the standard ZnONP diffraction spectrum (JCPDS: 89–0510) [58], the crystalline ZnONP was synthesized with the NSP support. The particles on NSP exhibited similar peaks to the pristine ZnONP, at 2*θ* = 32, 34, and 36° corresponding to (1 0 0), (0 0 2), and (1 0 1), which reflect the presence of the wurtzite phase of ZnONP. TEM was utilized to evaluate the supporting ability of NSP. As shown in Figure 3, hybrids of ZnONP on clay (*w*/*w* = 15/85) were found to have less of a tendency toward self-aggregation into large particles compared to the pristine ZnONP (Figure 3a). In Figure 3b,c, ZnONPs were well dispersed due to the strong interactions of ionic charges on the clay surface. Moreover, the usage of NSP as a support led to a better particle dispersion than that obtained using MMT. This could be attributed to the exfoliated NSP having a higher surface-to-weight ratio. The supporting effect of NSP was further explained by the analysis of ZnONP/NSP at various weight ratios. As shown in Figure 4, TEM images implied that NSP showed limitations when supporting ZnONP; the maximal weight ratio of ZnONP/NSP was 15/85. Beyond this weight ratio, free ZnONPs would easily aggregate to large clusters, for example, at the weight ratio of ZnONP/NSP (30/70) (Figure 4c). The average particle size (80.5 ± 24.0 nm) and the size distribution of the 15/85 hybrid are shown in Figure 4b.

### 2.2. Synthesis of AgNP/NSP and AgNP/ZnONP/NSP Hybrids

The NSP-supported AgNP (AgNP/NSP) was prepared according to previously reported procedures [50]. The AgNP/NSP formation process, illustrated in (Figure 1c), could be monitored by the absorption peak in the UV-Vis spectrum. The peak height at 411 nm indicated the concentration of AgNPs in aqueous suspension corresponding to the gradual conversion of silver nitrate, as shown in Appendix A. Compositions of AgNP/NSPs at various weight ratios were analyzed by TEM, as shown in Appendix A. The hybrid of AgNP/NSP (1/99) with a small particle size and fine dispersion in aqueous solutions was selected for the optimized sample to study the details of cytotoxicity and antibacterial activity. The AgNP/ZnONP/NSP hybrid was prepared by a two-step process, as shown in Figure 1d. Both weight ratios with 1/5/99 and 1/10/99 of AgNP/ZnONP/NSP were characterized by UV-visible spectroscopy. In Figure 5, it was shown the AgNP could be successfully synthesized onto the NSP in the presence of ZnONP without any interference. According to those absorption peaks of AgNP/ZnONP/NSP, no significant peak shift was observed compared to that of AgNP/NSP. Hybrids of AgNP/ZnO/NSP (1/10/99) were further analyzed by TEM, as shown in Figure 6, with a size in the range of 3 to 25 nm; AgNP and ZnONP are indicated by red arrows and white arrows, respectively. Referring to a previous study [42] and Appendix A, the size of AgNP was recorded as ≤5 nm.

### 2.3. Cytotoxicity of the Hybrids

To address the safety issue, hybrids of AgNP/NSP (1/99, *w*/*w*) and AgNP/ZnONP/NSP (1/5/99 and 1/10/99, *w*/*w*/*w*) were evaluated for cytotoxicity according to ISO 10993-5. As shown in Figure 7, all hybrids were found to be nontoxic toward the fibroblasts at 10 ppm, which was the effective concentration for the inhibition of the growth of *E. coli* (Table 1). In addition, the toxicological effect was reduced by introducing ZnONP to Ag at a 10:1 weight ratio. The cell viability was 70% for treatment with AgNP/ZnONP/NSP (1/10/99) at 100 ppm, whereas significantly lower cell viabilities were shown for the treatment of AgNP/NSP (1/99) and AgNP/ZnONP/NSP (1/5/99) at the same concentration. Regarding the cytotoxicity of Ag, other studies have controlled particle size [59] and green synthesis [60]. Our studies indicated that incorporating ZnONP into AgNP/NSP could significantly reduce the cytotoxicity of AgNP/NSP. The reason for this might be that the amount of Ag in AgNP/ZnONP/NSP was lower than that in AgNP/NSP, as ZnONP demonstrated much lower cytotoxicity when compared to AgNP [54,55]. However, further investigations are required to determine the optimal composition of AgNP/ZnONP/NSP and to verify the mechanism of this; the cellular ROS pathway and metal ions release have been reported as the two main factors affecting cytotoxicity [61,62].

### 2.4. Antibacterial Efficacies of the Hybrids

Different kinds of combinations of Ag and ZnO have been reported to achieve synergistic antibacterial activities [63,64,65,66,67,68] and applications, such as Ti implants [69], dressing [70], and biofilm impeders [71]. In this study, our uniquely NSP-supported ZnONP and NSP-supported AgNP/ZnONP were evaluated for their antibacterial efficacies using the standard micro-dilution method. As summarized in Table 1, ZnONP/NSP (15/85) showed weaker antibacterial efficacies than AgNP/NSP. Previous reports [72,73] revealed that Gram-positive bacteria (*S. aureus*) were more sensitive to exposure to ZnO in comparison to Gram-negative bacteria (*E. coli*). This result was in accordance with antibacterial tests of ZnONP/NSP. It was noteworthy that AgNP/ZnONP/NSP (minimum bactericidal concentration (MBC): 400 ppm) exhibited much stronger antibacterial activities against *S. aureus* than AgNP/NSP (MBC: 1500); AgNP/ZnONP/NSP showed an enhancement of antibacterial efficacy against *S. aureus* by approximately four times compared to AgNP/NSP. This result implies that ZnONP serves as a promoter in Ag/NSP regarding antibacterial behaviors. This synergistic effect might be attributable to the H_2_O_2_ produced from ZnONP [74]. Another explanation might be that the immobilized particles can stabilize silicate platelets, preventing them from forming precipitates, owing to the bulky particles on the NSP. According to the previous study, NSPs exhibited unique antibacterial activities caused by physical capturing mechanisms due to the platelets adhering to microbe surfaces [32]. However, the intensive interaction between aggregated NSPs caused the adverse condition of bacteria adherence. The in situ synthetization of nanoparticles on the silica surface was the method used to modify NSPs to improve their dispersity. Due to the synthetization of bulky particles on NSPs, the contact area and self-interaction of NSPs would reduce. Then, the particles-on-platelet nanohybrids could be well-dispersed in the water medium, as shown in Appendix A; similar concepts have been proposed in previous studies [75,76]. Furthermore, the probability of collision between particles-on-platelet nanohybrids and bacteria would be enhanced.

Moreover, the efficacy of bacterial growth inhibition was also evaluated. In Appendix A, it was shown that the growth of *S. aureus* could be entirely inhibited from contacting the MBC of AgNP/ZnONP/NSP for 3 h, in contrast to the incomplete inhibition of AgNP/NSP for 3 h at its MBC. Overall, the results indicated that the incorporation of ZnONP could improve the antibacterial efficacy of AgNP/NSP against Gram-positive *S. aureus*, which might resist Ag due to its thick and negative charge peptidoglycan layer (~30 nm) [77].

## 3. Materials and Methods

### 3.1. Materials

By using NSP as the support, the hybrid of ZnONP and AgNP was synthesized at different weight compositions. The NSP was prepared by the exfoliation process from the Na^+^–MMT (Nanocor Co., Aberdeen, MS, USA). As shown in (Figure 1a), the NSPs have the geometric shape of a platelet with dimensions of ca. 100 × 100 × 1 nm^3^ and the cationic exchange capacity (CEC) of 120 cmol (+)/kg. Zinc acetate (Zn(CH_3_COO)_2_·2H_2_O) was purchased from Alfa Aesar Co., Ward Hill, MA, USA; sodium hydroxide (NaOH) was purchased from SHOWA Co., Seoul, Republic of Korea; silver nitrate (AgNO_3_) was purchased from J. T. Baker, Inc., Phillipsburg, NY, USA; and sodium borohydride (NaBH_4_) was purchased from Acros Organics, Geel, Belgium.

### 3.2. Synthesis of ZnONP/NSP Hybrids

ZnONP/NSP hybrids were prepared at the composition weight ratios of 7/93, 15/85, and 30/70. A typical ZnONP procedure at the weight ratio of 15/85 is exemplified in the following. A three-necked and round-bottomed flask with a mechanical stirring, a reflux condenser, a heating mantle, and a nitrogen inlet–outlet line were charged with NSP in a water slurry (207.1 g, 1.2 wt% in deionized water). NSP suspension was stirred well by a mechanical stirrer at 500 rpm for 0.5 h before adding the Zn(CH_3_COO)_2_·2H_2_O solution (24.3 g, 5.0 wt% in deionized water). The mixture was then heated at 90 °C for 0.5 h, followed by the drop-wise addition of NaOH aqueous solution (66 g, 1.0 wt% in deionized water). The mixture was heated and refluxed under nitrogen for 1 h while stirring at 90 °C. The crude product was filtered and washed with deionized water several times. The formation of ZnONP/NSP solution was examined by UV-Vis spectrophotometer and X-ray powder diffractometer. Particle sizes were estimated by a transmission electron microscope (TEM).

### 3.3. Synthesis of AgNP/NSP and AgNP/ZnONP/NSP Hybrids

AgNP/NSP was prepared by following the example of the previous publication [42]. The AgNP/ZnONP/NSP hybrids were synthesized by a two-step process involving an AgNP in situ reduction from AgNO_3_ followed by immobilization on ZnONP/NSP. Bimetallic AgNP/ZnONP/NSP hybrids were prepared at the composition weight ratios of 1/5/99 and 1/10/99. The process for 1/5/99 composition was described as follows. NSP in water slurry (60 g, 5.0 wt% in deionized water) was charged to a three-necked and round-bottomed flask equipped with a mechanical stirring, a reflux condenser, a heating mantle, and a nitrogen inlet–outlet line. The NSP suspension was stirred well by a mechanical stirrer at 500 rpm for 0.5 h. The Zn(CH3COO)_2_·2H_2_O solution (8.09 g, 5.0 wt% in deionized water) was then added, and the mixture was maintained at 90 °C for 0.5 h. An aqueous solution of NaOH (21 g, 1.0 wt% in deionized water) was then added in a drop-wise manner and the mixture was heated to refluxing temperature under nitrogen for 1 h while stirring continuously. The crude product was filtered and washed with deionized water several times. The formation of ZnONP/NSP (5/99, *w*/*w*) suspension (100 g, 2.0 wt% in deionized water) was further charged to a round-bottomed flask equipped with mechanical stirring. Subsequently, the suspension was mixed by a mechanical stirrer at 500 rpm for 0.5 h before adding AgNO_3_ solution (3.1 g, 1.0 wt% in deionized water). An aqueous solution of NaBH_4_ (0.3 g, 1.0 wt% in deionized water) was then added. The mixture was then stirred for 1 h. The mixture changed from yellow to brown, indicating the reduction from Ag^+^ to Ag^0^ in the presence of the NaBH_4_ reducing agent. The formation of AgNP/ZnONP/NSP suspension was characterized by a UV-Vis spectrophotometer (Hitachi U-4100, Tokyo, Japan) and the particle size was estimated by a transmission electron microscope (TEM).

### 3.4. Instrumentation and Analyses

Ultraviolet–visible spectroscopy (UV-Vis) absorptions of hybrids were performed with a Hitachi U-4100 spectrophotometer. Crystal structures were determined using X-ray powder diffraction (XRD) performed on a Schimadzu SD-D1 diffractometer with a Cu target (λ = 1.5418 Å) at a generator voltage of 35 kV, a generator current of 40 mA, and a scanning rate of 2° min^−1^. Particle sizes of the hybrid were measured by transmission electron microscope (TEM, JEOL JEM-1230, JEOL Co., Ltd., Tokyo, Japan) operating at 100 kV with a Dual Vision CCD camera (Gatan, Inc., Pleasanton, CA, USA), and the size distribution was estimated from 100 individual particles.

### 3.5. Cytotoxicity Tests

An in vitro cytotoxicity test was performed following the modified method of ISO 10993-5. CCL 163 (Balb/3T3 clone A31), the mouse fibroblast cell line, obtained from American Type Culture Collection, was seeded in 24-well tissue culture plates at 5 × 10^4^ cells per well. After 24 h incubation, the medium was removed and replaced by a test medium containing 0.1 mL of the tested sample (suspension of the Ag-containing materials) and 0.9 mL of medium. The final sample concentrations for direct contact were 1, 10, and 100 ppm. After 24 h treatment, 3-(4,5-dimethylthiazol-2-yl)-2,5-diphenyl tetrazolium bromide (MTT) solution was added to each well (100 μg/well) and incubated at 37 °C for 2 h. The supernatant was removed and the dimethyl sulfoxide (DMSO) was added to each well to dissolve the formazan crystal. Afterward, the supernatant solution from each well was transferred to a 96-well plate. The spectrophotometric absorbance at 570 nm was measured by a multi-well ELISA reader to evaluate cell viability. All measurements were performed in six repeated wells and were performed independently in triplicate.

### 3.6. Antibacterial Tests

The antibacterial activities of ZnONP/NSP, AgNP/NSP, and AgNP/ZnONP/NSP were determined using the standard micro-dilution method. Bacterial strains, including Gram-negative Escherichia coli (DH5α) and Gram-positive Staphylococcus aureus, were obtained from Super Laboratory Co. (New Taipei, Taiwan). The minimum bactericidal concentration (MBC) is defined as the lowest concentration for an antimicrobial agent that is bactericidal to ≥99.9% of an original inoculum to prevent the growth of a microorganism after subculture onto the antibiotic-free media. The inhibition of bacterial growth was measured following the National Committee for Clinical Laboratory Standards. Typically, bacteria were cultured overnight in Luria−Bertani (LB) broth. Then, 100 μL of the suspension was inoculated in fresh medium to restart the cell cycle. After incubation at 37 °C for 3 h, cells were synchronized at the log phase of the growth curve, with the optical density at 600 nm (OD. 600) of 0.3−0.5. The designed concentration of test materials was added to the bacterial suspension with 1 × 10^6^ colony formation unit per milliliter (CFU mL^−1^). At 0, 3, and 24 h, portions of the culture were taken out to spread onto Luria−Bertani (LB) agar plates for another 24 h incubation at 37 °C. The number of colonies was counted to determine the antibacterial efficacy.

## 4. Conclusions

Nanoscale silicate platelet (NSP) demonstrated an advantageous supporting effect for ZnONP and AgNP. The large surface-to-weight ratio of NSP can facilitate the immobilization of the nanoparticles via van der Waals force during and after the in situ formation. AgNP/ZnONP/NSP exhibited improved antibacterial efficacy compared to ZnONP/NSP and demonstrated about four times enhanced antibacterial efficacy compared to AgNP/NSP against *S. aureus*. The final design of AgNP/ZnONP/NSP (1/10/99) further exhibited better biocompatibility than AgNP/NSP (1/99) regarding cytotoxicity at high concentration (100 ppm). This AgNP/ZnONP/NSP, with its advantageous properties, can be considered for use as a potential coating material on food packaging, as well as for medical uses.

## Data Availability

Not applicable.

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
