# Peer review of "Synergistic Effects of Silicate-Platelet Supporting Ag and ZnO, Offering High Antibacterial Activity and Low Cytotoxicity"

_ijms, 2023, doi:10.3390/ijms24087024_

Round 1
Reviewer 1 Report
The manuscript entitled “Synergistic Effect of Silicate-Platelet Supporting Ag and ZnO for High Antibacterial Activity and Low Cytotoxicity” investigate the nanoscale silicate platelet as a support for ZnO and Ag nanoparticles.
The novelty of the presented paper relies on the hybrid synthesis of Ag/ZnO/NSP but there is similar and more detailed research that has already been published in 2011. Reference No.36, and in 2020. https://www.mdpi.com/2073-4360/12/2/482.
Nevertheless, this manuscript has great potential if the research focuses on the topic indicated in the last sentence of the conclusion.
Also, further investigation is requested:
1. Analysis of nanohybrids: concentration of Ag and Zn in material (ICP-OES analysis, EDS).
2. Topography of nanohybrids. Show in TEM figures the locations of Ag and ZnO nanoparticles.
3. Stability of materials.
4. Propose a possible mechanism of a synergistic effect of nanohybrids
Author Response
Reviewer -1
Point 1. Analysis of nanohybrids: concentration of Ag and Zn in material (ICP-OES analysis, EDS).
Response 1.
According to the preparation of AgNP/ZnONP/NSP (1/5/99), as mentioned in the section of “Materials and Methods” of this article, the weight of Ag and Zn were calculated as bellowed:
Zn(CH3COO)2·2H2O MW: 219.51
Zn MW: 65.4
AgNO3 MW: 169.87
Ag MW: 107.8
Zn (g) à 100*2%*8.09*5%*65/219.51/(60*5%+8.09*5%*65/219.51) = 0.07679 (g)
Ag (g) à 3.1 x 1% x 107.8/169.87 = 0.01967 (g)
Point 2. Topography of nanohybrids. Show in TEM figures the locations of Ag and ZnO nanoparticles.
Response 2.
Thanks for your suggestion. We have pointed out the locations of Ag and ZnO in Figure 6.
Figure 6. TEM image of AgNP/ZnONP/NSP with a weight ratio of 1/10/99. The size distribution is inserted. AgNP and ZnONP are indicated by red arrows and white arrows, respectively.
Point 3. Stability of materials.
Response 3.
Thank you for the comment. Due to the presence of nano silicate platelets NSP as a supporting effect, the synthesized Ag/ZnO/NSP is stable against oxygen oxidation and thermal degradation. This is evidenced by the maintenance of the characteristic AgNP golden color over months during the storage at room temperature in vials.
Point 4. Propose a possible mechanism of a synergistic effect of nanohybrids
Response 4.
Thank you for the comment. The high efficacy for antibacterial activity is postulated. Since the NSP is possessing a nanometer-thick platelet geometric shape (ca. 1 x 100 x 100 nm in dimension), it high surface area can render van der Waals force for associating Ag and ZnO nanoparticles together in one NSP until. As a result, the Ag/ZnO/NSP nanohybrids individually can demonstrate a “high local concentration” in terms of the overall concentration in effect for interacting with bacterial surface. In other words, the bacteria were under attack by a higher “apparent” or local concentration of Ag or ZnO nanoparticles.

Reviewer 2 Report
The manuscript " Synergistic Effect of Silicate-Platelet Supporting Ag and ZnO 2 for High Antibacterial Activity and Low Cytotoxicity" in this paper, the authors reported the results on the co-synthesis of zinc oxide and silver nanoparticles on the nano-silicate platelet.
The obtained materials were investigated from an optical, structural and morphological point of view using experimental techniques such as UV-Vis, XRD, TEM, but also from a biological point of view, evaluating the cytotoxicity of the samples on the mouse fibroblast cell line. The antibacterial activity against E-coli and Staphylococcus aureus was also assessed. The authors find that AgNP/ZnONP/NSP sample exhibited a more antibacterial efficacy than ZnONP/NSP sample. The effect is of about four times enhanced in the case of sample AgNP/ZnONP/NSP than in the case of sample AgNP/NSP. Moreover, the sample AgNP/ZnONP/NSP (1/10/99) presents a better biocompatibility than the sample AgNP/NSP (1/99), at high concentration (100 ppm), cell viability is higher in the case of the AgNP/ZnONP/NSP (1/10/99) compared to the sample AgNP/NSP (1/99).
The content of this study is interesting and can be published in International Journal of Molecular Sciences in the present form.
Author Response
Thanks for the comment. We have sent our manuscript to MDPI for English editing. English editing ID: English-63765

Reviewer 3 Report
Upon reviewing your manuscript, I find your work interesting, but I do not believe it can be published in its current form. Therefore, some revisions must be done so that it may be published in this journal. I have the following comments and recommendations:
- The abstract is qualitative; authors should use this section to present the main results in a qualified way. In this sense, highlight the novelty of the study.
- The introduction of relevant background and research progress was not comprehensive enough.
- The doping can improve several material properties and the choice for doping depends on several factors. With respect to this, the authors do not justify the doping concentrations used in this system. Due to the importance for this system, a reasoned explanation must be attached as part of the motivation and justification of the work.
- In the experimental section some details should be included. Example:
The purity of the raw materials used must be included.
What is the pH of the solution at the start and end of the synthesis?
- About the study by XRD, there are several points that must be clarified.
The crystallographic record for the plans shown must be included in the text.
Important structural parameters in doped systems, such as: crystallite size and lattice constants, must be calculated and discussed.
There are several diffraction peaks that have not been identified. Do a quantitative analysis of XRD patterns. Evaluate the influence of doping on the structure. This is important and will obviously have effects on the properties analyzed in this study.
- The optical energy gap must be calculated from the UV-Vis absorption spectra. Evaluate the influence of Ag on this important optical parameter.
- Porosity analysis will be important to assess its effects on properties and antibacterial tests
- In its present form, none of the characterization techniques show that Ag is present in this study. EDS or XPS will be required in addition to providing quantified information on XRD patterns.
- The discussion about: Cytotoxicity of the hybrids is qualitative and some hypothetical cases, stronger discussions and based on the literature must be made to quantify the results.
Author Response
Reviewer -3
Point 1. The abstract is qualitative; authors should use this section to present the main results in a qualified way. In this sense, highlight the novelty of the study.
Response 1.
Thank you for the comment. We have revised the abstract and main result to highlight the novelty of the study as the reviewer’s suggestion. The revised abstract is shown as follows:
Silver nanoparticles (AgNPs) are remarkably able to eliminate microorganisms but induce cytotoxicity in mammalian cells, and zinc oxide nanoparticles (ZnONPs) is considered to have a wide bactericidal effect with weak cytotoxicity. In this study, both zinc oxide nanoparticles and the silver nanoparticles were co-synthesized on a nano-silicate platelet (NSP) to prepare a hybrid of AgNP/ZnONP/NSP. Ultraviolet–visible spectroscopy (UV-Vis), X-ray diffraction (XRD), and transmission electron microscopy (TEM) were used to characterize the formation of nanoparticles on the NSP. Synthesized ZnONP/NSP (ZnONP on NSP) was confirmed by the absorption peaks on UV-Vis and XRD. AgNP synthesized on ZnONP/NSP was also characterized by UV-Vis, and ZnONP/NSP showed no interference with synthesis. The images of TEM demonstrated that NSP provides physical support for the growth of nanoparticles, could prevent the inherent aggregation of ZnONP. In antibacterial tests, AgNP/ZnONP/NSP exhibited more efficacy against Staphylococcus aureus (S. aureus) than ZnONP/NSP (ZnONP was synthesized on NSP) and AgNP/NSP (AgNP was synthesized on NSP). In cell culture tests, 1/10/99 (weight ratio) of AgNP/ZnONP/NSP exhibited low cytotoxicity for mammalian cells (>100 ppm). Therefore, AgNP/ZnONP/NSP, containing both AgNP and ZnONP, with both strong antibacterial qualities and low cytotoxicity, showed potentially advantageous medical utilizations due to its antibacterial properties.
Point 2. The introduction of relevant background and research progress was not comprehensive enough.
Response 2.
Thank you for the suggestion. We have added more relevant background and research progress in the section introduction. The revised section of the introduction is shown as follows:
- Na+-MMT, a bentonite clay, with 1.20 mequiv/g of cationic exchange capacity (CEC) or exchangeable Na+ capacity, is generally composed of 2/1 layered silicate/aluminum oxides, geometrically arranged in two tetrahedral sheets sandwiching an edge-shared octahedral sheet [27,28].
- NSP has been revealed to restrain the proliferation and growth of bacteria in previous studies. The surface charge of NSPs, attributed to SiO-Na+, results in significant antimicrobial activities; platelets adhere to the surface of microbes due to their physical capturing mechanisms [31, 32]. Moreover, surfactant-modified NSP has exhibited broad antiviral activities [33] and the potential to treat gray mold disease in strawberries [34].
- Silver has been applied in food protection and wound treatment, such as burning, for several years [43, 44]. Numerous silver compounds, such as silver oxide, silver powder, silver chloride, and silver nitrate, are used as antimicrobial materials [45-47]. However, the issue of toxicity should be considered when using silver-based compounds as anti-microbial agents due to the cytotoxicity and genotoxicity in human cells shown in the previous study [48].
- The particle size of AgNP and the antibacterial activity of AgNP/NSP were affected by the weight ratio of Ag and NSP [50]. In addition, AgNP/NSP demonstrated low cyto-toxicity and further promoted wound healing [37].
- ZnONP, which is non-toxic in microorganisms and biocompatible in human cells, has been widely used as an antibacterial agent due to its antibacterial activities, for exam-ple, in food packaging, to inhibit the growth of foodborne pathogens. Four antibacterial mechanisms of ZnONP were proposed: contact with cell walls and ZnONP, destruct-ing the integrity of cells, Zn2+ ion liberation, and reactive oxygen species (ROS) for-mation [57].
- In the present study, AgNP/ZnONP/NSP was prepared using a two-step process, meaning that ZnONP/NSP was initially formed by the in situ dehydration of Zn(OH)2 on NSP, and AgNP was then deposited in the presence of the reducing agent (NaBH4) and AgNO3 solution. X-ray diffraction (XRD) was utilized to confirm the synthesis of ZnONP on NSP, and NSP seemed to prevent the self-aggregation of ZnONP. AgNP was characterized by ultraviolet–visible spectroscopy (UV-Vis) to ensure the for-mation of AgNP/ZnONP/NSP. A transmission electron microscope (TEM) was used to catch the morphology and determine the diameter of ZnONP, ZnONP/NSP, and AgNP/ZnONP/NSP. AgNP/ZnONP/NSP (weight ratio: 1/10/99) demonstrated low cy-totoxicity when incubated with fibroblasts, based on ISO 10993-5. In addition, AgNP/ZnONP/NSP showed much stronger antibacterial activities against S. aureus and E. coli than ZnONP/NSP and AgNP/NSP. The hybrid, AgNP/ZnONP/NSP, has shown high potential in clinical applications and the food industry.
Point 3. The doping can improve several material properties and the choice for doping depends on several factors. With respect to this, the authors do not justify the doping concentrations used in this system. Due to the importance for this system, a reasoned explanation must be attached as part of the motivation and justification of the work.
Response 3.
Thank you. Besides the chemistry issues, the NSP as the support can provide a new role of “physical” effect. Specifically, as mentioned above, the NSP geometric shape of nanometer-thin platelets can render physical adsorption through van der Waals force and adhered by Ag and ZnO nanoparticles showing stability and local-concentration effect on bacterial surface.
Point 4. In the experimental section some details should be included. Example:
Point 4-1. The purity of the raw materials used must be included.
Response 4-1.
Thank you for the comment. We have added the purity of the raw materials as mentioned in the section of “Materials and Methods” of this article. Zn(CH3COO)2·2H2O (Alfa Aesar, 11559.30), ACS, 98.0-101.0%.
AgNO3 (J. T. Baker, 1182.0025), ACS, min. 99.9 %
NaBH4 (Acros Organics, 200055000), 99%
Point 4-2. What is the pH of the solution at the start and end of the synthesis?
Response 4-2.
Thank you. NaOH was used for the reaction, and the products were washed by water to remove the basicity.
Point 4-3. About the study by XRD, there are several points that must be clarified.
The crystallographic record for the plans shown must be included in the text.
Important structural parameters in doped systems, such as: crystallite size and lattice constants, must be calculated and discussed.
There are several diffraction peaks that have not been identified. Do a quantitative analysis of XRD patterns. Evaluate the influence of doping on the structure. This is important and will obviously have effects on the properties analyzed in this study.
Response 4-3.
Thank you for the comment. The major diffraction peaks of XRD have been identified in the data. In our work, the crystalline lattice of XRD is less relevant since the antimicrobial efficacy of high dilution of these materials was the main theme of this paper.
Point 4-4. The optical energy gap must be calculated from the UV-Vis absorption spectra. Evaluate the influence of Ag on this important optical parameter.
Point 4-5. Porosity analysis will be important to assess its effects on properties and antibacterial tests
Response 4-4. & 4-5.
Thanks for the opinion. The details of UV-Vis absorption spectra may not the direct concern in this study, and the Porosity analysis may not the major properties affected the antibacterial abilities of metal nanoparticles. The emphasis of this paper is the biofunction rather than the details of the analytical works.
Point 4-6. In its present form, none of the characterization techniques show that Ag is present in this study. EDS or XPS will be required in addition to providing quantified information on XRD patterns.
Response 4-6.
Thanks for the opinion. Figure 5. showed AgNP was successfully synthesized on the NSP. It was shown the AgNP could be successfully synthesized onto the NSP in the presence of ZnONP without any interference. According to those absorption peaks of AgNP/ZnONP/NSP, no significant peak shift was observed compared to that of AgNP/NSP.
Point 4-7. The discussion about: Cytotoxicity of the hybrids is qualitative and some hypothetical cases, stronger discussions and based on the literature must be made to quantify the results.
Response 4-7.
Thanks for the suggestion. We have made more discussions in section 2.3. The reason for this might be that the amount of Ag in AgNP/ZnONP/NSP was lower than that in AgNP/NSP, as ZnONP demonstrated much lower cytotoxicity when compared to AgNP [54,55].

Round 2
Reviewer 1 Report
There is an error in the figure numbers in the Supplementary file. Please correct
Author Response
Point 1. There is an error in the figure numbers in the Supplementary file. Please correct.
Response. Thanks for your suggestion. We have corrected both the supplementary file and Supplemental Materials in the manuscript.

Reviewer 3 Report
This version may be considered for publication.
Author Response
Point 1. This version may be considered for publication.
Response. Thanks for your comment.